# Clinical Characterization of Alagille Syndrome in Patients with Cholestatic Liver Disease

**DOI:** 10.3390/ijms241411758

**Published:** 2023-07-21

**Authors:** Natalia Semenova, Elena Kamenets, Eleonora Annenkova, Andrey Marakhonov, Elena Gusarova, Nina Demina, Daria Guseva, Inga Anisimova, Anna Degtyareva, Natalia Taran, Tatiana Strokova, Ekaterina Zakharova

**Affiliations:** 1Research Centre for Medical Genetics, 115522 Moscow, Russia; elenakamenec@yandex.ru (E.K.); ann.eleonora12@gmail.com (E.A.); marakhonov@gmail.com (A.M.); elenaleksgus@gmail.com (E.G.); demina@med-gen.ru (N.D.); guseva@med-gen.ru (D.G.); anisimova-inga@med-gen.ru (I.A.); doctor.zakharova@gmail.com (E.Z.); 2National Medical Research Center for Obstetrics, Gynecology and Perinatology named after V.I. Kulakov, Ministry of Health of the Russian Federation, 115522 Moscow, Russia; mishel_doc@mail.ru; 3Department of Neonatology, First Moscow State Medical University named after I.M. Sechenov, 115522 Moscow, Russia; 4Federal Research Centre of Nutrition and Biotechnology, 115522 Moscow, Russia; pknt@mail.ru (N.T.); strokova_t.v@mail.ru (T.S.)

**Keywords:** Alagille syndrome, cholestasis, biochemical characteristics, *JAG1* gene

## Abstract

Alagille syndrome (ALGS) is a multisystem condition characterized by cholestasis and bile duct paucity on liver biopsy and variable involvement of the heart, skeleton, eyes, kidneys, and face and caused by pathogenic variants in the *JAG1* or *NOTCH2* gene. The variable expressivity of the clinical phenotype and the lack of genotype–phenotype correlations lead to significant diagnostic difficulties. Here we present an analysis of 18 patients with cholestasis who were diagnosed with ALGS. We used an NGS panel targeting coding exons of 52 genes, including the *JAG1* and *NOTCH2* genes. Sanger sequencing was used to verify the mutation in the affected individuals and family members. The specific facial phenotype was seen in 16/18 (88.9%). Heart defects were seen in 8/18 (44.4%) patients (pulmonary stenosis in 7/8). Butterfly vertebrae were seen in 5/14 (35.7%) patients. Renal involvement was detected in 2/18 (11.1%) cases—one patient had renal cysts, and one had obstructive hydronephrosis. An ophthalmology examination was performed on 12 children, and only one had posterior embryotoxon (8.3%). A percutaneous liver biopsy was performed in nine cases. Bile duct paucity was detected in six/nine cases (66.7%). Two patients required liver transplantation because of cirrhosis. We identified nine novel variants in the *JAG1* gene—eight frameshift variants (c.1619_1622dupGCTA (p.Tyr541X), c.1160delG (p.Gly387fs), c.964dupT (p.C322fs), c.120delG (p.L40fs), c.1984dupG (p.Ala662Glyfs), c.3168_3169delAG (p.R1056Sfs*51), c.2688delG (p.896CysfsTer49), c.164dupG (p.Cys55fs)) and one missense variant, c.2806T > G (p.Cys936Gly). None of the patients presented with *NOTCH2* variants. In accordance with the classical criteria, only six patients could meet the diagnostic criteria in our cohort without genetic analysis. Genetic testing is important in the diagnosis of ALGS and can help differentiate it from other types of cholestasis.

## 1. Introduction

Alagille syndrome (ALGS) is an autosomal dominant, multisystem condition with variable phenotypic penetrance that was first described in 1969 by Dr. Daniel Alagille as an arteriohepatic dysplasia [1]. The prevalence was estimated at 1:30,000–1:50,000 [2]. The major clinical symptoms of ALGS are bile duct paucity on liver biopsy, cholestasis with a high level of gamma-glutamyl transferase (GGT), congenital cardiac defects (most commonly involving the pulmonary arteries), butterfly vertebrae, ophthalmologic abnormalities (primarily posterior embryotoxon), and a characteristic facial phenotype. ALGS is caused by heterozygous pathogenic variants in one of two genes, mostly in *JAG1* (chromosome 20p12.2) and *NOTCH2* (chromosome 1p12-p11) [3]. Jagged-1 (JAG1) is a transmembrane ligand of the Notch signaling pathway. A partial loss of the Jagged-1 protein due to a heterozygous loss-of-function mutation of the *JAG1* gene leads to disruptions in the typical bile duct development, resulting in bile duct paucity [4]. Recent mutation analysis of patients with *JAG1* variants has identified mutations in all 26 exons of the gene. Haploinsufficiency caused by truncation or early transcriptional termination of *JAG1* is very common for ALGS patients [5]. Variants in the *NOTCH2* gene are significantly more rarely detected in ALGS patients, occurring in 2.5% of probands from a recent analysis. However, pathogenic variants in the *NOTCH2* gene are more likely to be missense (68%) than in the *JAG1* gene (15%) [5]. No genotype–phenotype correlations for *JAG1* or *NOTCH2* have been identified [6].

The cholestatic liver disorder is commonly presented in ALGS with a frequency of 89–100% [6]. Liver biopsy mostly shows the paucity of the intrahepatic bile ducts. In infants younger than three months, bile duct paucity is not always presented, and the liver biopsy may demonstrate ductal proliferation, resulting in a possible misdiagnosis of ALGS as biliary atresia. Moreover, bile duct paucity is not pathognomonic of ALGS and can be found in various genetic disorders (Down syndrome), metabolic disorders (α_1_-antitrypsin deficiency), infections (congenital cytomegalovirus), and others. It is important to differentiate ALGS from other causes of cholestasis, particularly biliary atresia, because the Kasai portoenterostomy may worsen the prognosis of ALGS patients [7]. New therapeutic approaches are currently under investigation, including bile acid transporter inhibitors and others, which could be effective in ALGS [8]. A summation of clinical and genetic data can help in the diagnosis and appropriate management of these patients.

The purpose of this study is to establish the clinical presentation and biochemical characteristics in a group of patients with ALGS with cholestasis. 

## 2. Results

### 2.1. Clinical Evaluation

Eighteen patients were included in the study (12 boys and 6 girls). Two children (8 and 9 years) are siblings—a boy and his older sister (Table 1). The sister was diagnosed after her brother at 6 years of age; she had not had any symptoms and was thought to be healthy. The age of onset in the cohort varied from 5 days to 24 months (mostly in the first month of life). The median age at the time of diagnosis was 21 months (range: 2 to 142 months): seven children were under the age of 6 months; seven were between 1 year and 2 years; and four were older than 2 years of age. 

Clinical features are summarized in Table 1. Growth was impaired in most children with ALGS. Medians of height and weight z-scores were low and amounted to −1.8 (range: 0.8 to −3.48) and −1.9 (range: 0.55 to −3.49), respectively.

A specific phenotype (a broad forehead, deep-set eyes, up-slanting palpebral fissures, prominent ears, a straight nose with a bulbous tip, and a pointed chin) was seen in 16/18 (88.9%); five of them were under 1 year of age (5/16) (Figure 1 and Figure 2). The two children without specific facial features were 2 months old. Eight/fifteen were shown to have pruritus, representing 53.3% of the study cohort, and only one of them was under 1 year of age. Heart defects were seen in 8/18 (44.4%) patients (pulmonary stenosis in 7/8). Butterfly vertebrae were observed in 5/14 (35.7%) patients. Other reported abnormalities, including clinodactyly of the fifth finger, shortened distal phalanges, and radio-ulnar synostosis, were not noted in our cohort. Renal involvement was detected in 2/18 (11.1%): one patient had renal cysts, and one had obstructive hydronephrosis. 

An ophthalmology examination was performed on 12 children, and only one had posterior embryotoxon (8.3%). A percutaneous liver biopsy was performed in nine cases. Characteristic features of paucity of interlobular bile ducts were detected in six/nine cases (66.7%). Two patients required liver transplantation because of cirrhosis.

All patients had cholestasis with a high level of GGT. In all children, biochemical characterization of the patients showed elevation of the following: ALAT with a mean of 215 ± 170 U/L; ASAT with a mean of 227 ± 144 U/L; and GGT levels with a mean of 373 ± 245 U/L. Total bilirubin (TB) was elevated in 14/18. The mean of maximal elevations of TB was 109 ± 116 µmol/L, and the mean of maximum maximal elevations of total cholesterol was 6.3 ± 1.9 mmol/L (Figure 2A). 

### 2.2. Gene Variation in ALGS

The *JAG1* gene variants of ALGS in our subjects are shown in Table 1. We identified nine novel variants (Figure 3A). These variants were not present in the Human Gene Mutation Database (HGMD) v. 2022.1 or ClinVar [19]. All of them were absent from the gnomAD database [20]. These included eight frameshift variants: c.1619_1622dupGCTA (p.Tyr541X); c.1160delG (p.Gly387fs); c.964dupT (p.C322fs); c.120delG (p.L40fs); c.1984dupG (p.Ala662Glyfs); c.3168_3169delAG (p.R1056Sfs*51); c.2688delG (p.896CysfsTer49); and c.164dupG (p.Cys55fs). Null variants in the *JAG1* gene are predicted to cause nonsense-mediated mRNA decay. Taken together with the absence of the genomes and family background in gnomAD, all these variants are considered as pathogenic or likely pathogenic. One missense variant, c.2806T > G (p.Cys936Gly), is novel, but there is an alternative missense variant in 936 codons assessed as pathogenic, and the found variant is considered as likely pathogenic. Various predictions of this variant are listed in Appendix A. The range of variants identified in the cohort is shown in Figure 3B, with a preponderance of protein-truncating mutations. None of the patients presented with *NOTCH2* variants. 

All variants were confirmed by Sanger sequencing. Thirteen variants were de novo. In case 2, the variant was inherited from an unaffected mother, and three of the proband’s siblings did not have this variant or any symptoms of ALGS. In case 5, the variant was inherited from the unaffected mother, and the proband’s older sibling died after liver transplantation at 19 months of age; his genetic status is unknown. Two symptomatic siblings (8 and 9 years) from the same family inherited the previously described pathogenic *JAG1* variant from their unaffected mother. One child (case 11) was adopted, and we could not test his parents.

## 3. Discussion

Alagille syndrome is an autosomal dominant, multisystem disorder with a broad phenotypic presentation. The major clinical symptom of ALGS is a cholestatic liver disorder with a frequency of 89–100% [6]. Other common features, such as congenital cardiac defects (most commonly involving the pulmonary arteries), butterfly vertebrae, ophthalmologic abnormalities (primarily posterior embryotoxon), and a characteristic facial phenotype, are manifested with varying occurrences. ALGS is one of the most common causes of cholestatic liver disorder during childhood [21], and very often, it is difficult to differentiate this condition from other cholestasis. 

Here we examined in detail the eighteen patients with cholestasis who were diagnosed as having ALGS. This study described a cohort of children with ALGS, all of whom had cholestasis. 

All patients had laboratory signs of a cholestatic liver disorder, but one patient did not have clinical symptoms, such as jaundice, pruritus, and others. 

Classically, ALGS should be suspected in individuals with bile duct paucity in association with three of the five criteria as follows: specifically, cholestasis; cardiac defect; skeletal abnormalities; ophthalmologic abnormalities (most commonly posterior embryotoxon); and characteristic facial features, or four of the five classical criteria in the absence of bile duct paucity (1). In accordance with the classical clinical criteria, only six patients (2, 3, 11, 13, 15, 17 in Table 1) could meet the diagnostic criteria in our cohort without genetic analysis. Moreover, a percutaneous needle liver biopsy could not be performed in all cases (9/18 in our study), and it is not always informative, especially in infants (6/9 in our study). In our cohort, the most common symptom of ALGS was the characteristic facial phenotype (88.9%). Only two children did not have specific facial features, and they were under 3 months of age. We suspect that the characteristic phenotype is the most common symptom in patients with ALGL and cholestasis, especially in children over a year of age. 

One of the most common symptoms of cholestasis in ALGS is intensive pruritus, which is among the worst features of any cholestatic liver disease. Usually, pruritus becomes apparent at around 3–4 months of age, causing skin lesions and sleeping difficulties. The prevalence of pruritus in ALGS is highest for patients up to 2 years of age and lower for older participants. In our cohort, pruritus was observed in 8/15 (53.3%) patients, and only one of them was under 1 year of age.

In our cohort, heart disease was identified in 8/18 (44.4%) patients; this is lower than in previously described ALGS cohorts (up to 94%) [6,7], but, as in the literature, almost all of them (7/8) had pulmonary artery stenosis or hypoplasia.

Another commonly reported anomaly in ALGS is butterfly vertebrae, seen in 33–87%, formed by an incomplete fusion of the anterior arch of vertebrae [6]. In our study cohort, butterfly vertebrae were observed in 5/14 patients (35.7%). However, they may be presented in other genetic syndromes, such as VACTERL-association, Kabuki syndrome, and others. This anomaly should be considered in the differential diagnosis of ALGS with other genetic conditions.

Previous studies of large cohorts reported that renal anomaly was one of the common clinical features of ALGS, with a prevalence of 23–39% [22,23]. In our study, renal involvement was detected only in 2/18 (11.1%) patients. 

Ophthalmology examination should also be helpful for meeting additional criteria of ALGS. In our cohort, only one (1/12) patient had posterior embryotoxon (8.3%), even though posterior embryotoxon is found in 56–95% of patients with ALGS [6]. 

In all patients, we performed a genetic analysis. In our study cohort, *JAG1* gene variants were detected, and the proportions of missense (6%), nonsense (12%), frameshift (59%), and splice site (23%) variants were similar to other large cohort reports [22]. We identified nine novel variants, eight frameshift variants and one missense variant. In our study, we identified a novel de novo likely pathogenic missense variant, c.2806T > G (p.(Cys936Gly), amino acid substitutions that resulted in the loss of cysteine. In a large cohort study, it was shown that *JAG1* missense variants involving the loss of cysteine are commonly associated with the disease [22]. Four causative variants identified in the cohort (Table 1) were inherited from healthy mothers. Incomplete penetrance is a well-known characteristic of ALGS, which is hypothetically explained by the presence of modifier genes in ok genetic background [8]. 

In a previous study of 25 patients with ALGS and jaundice beyond 2 years of age and gene deletions had poor prognosis [23]. However, the frameshift variant c.2122_2125 del (p.(Gln708ValfsTer34)) in the JAG1 gene was detected in three unrelated probands (de novo) in the patients with resolving jaundice after 2 years of age. In the literature review, there are 17 additional descriptions of patients with the same variant and different outcomes. Authors suspect the 4-bp deletion is a hotspot variant, and it needs further research to clarify this site as a mutation hotspot and its relationship with patient outcome. Our patient (#3) with the same variant (de novo) was under 2 years of age and had features affecting the liver, heart, vertebrae, and face.

None of the patients presented with *NOTCH2* variants.

## 4. Materials and Methods

### 4.1. Patients

Here we present a retrospective analysis of a comprehensive examination of eighteen patients with cholestasis who were diagnosed with ALGS at the Research Centre for Medical Genetics in Moscow and were treated at the Federal Research Centre of Nutrition and Biotechnology in Moscow and the National Medical Research Center for Obstetrics, Gynecology and Perinatology named after V.I. Kulakov, Ministry of Health of the Russian Federation. All patients were tested for common infections, such as TORCH, and for the detection of antibodies of hepatitis B and hepatitis C. In addition, the levels of amino acids and acylcarnitines in plasma were analyzed by the MC/MC technology in all cases to exclude common aminoacidopathies. 

### 4.2. Ethical Approval and Consent to Participate

The clinical and molecular genetic study was performed in accordance with the Declaration of Helsinki and was approved by the Institutional Review Board of the Research Centre for Medical Genetics, Moscow, Russia (approval number 11/23.11.2021), with written informed consent obtained from each participant and/or their legal representative as appropriate.

### 4.3. Liver Function Tests 

Peripheral fasting blood samples were received from 18 ALGS patients for liver function tests, including serum alanine aminotransferase (ALAT), aspartate aminotransferase (ASAT), total bilirubin (TB), direct bilirubin (DB), gamma-glutamyl transferase (GGT), alkaline phosphatase (AlcPh), albumin (ALB), total cholesterol (TC), and triglycerides (Tr). 

### 4.4. Other Accessory Examinations 

Liver and spleen ultrasonic and MRI examinations were performed on all the ALGS patients. 

Percutaneous puncture liver biopsy was performed in nine cases. 

### 4.5. DNA Analysis

Genomic DNA was extracted from whole blood with EDTA using the GeneJET Genomic DNA Purification Kit (Thermo Fisher Scientific, Waltham, MA, USA). Direct sequencing was performed on an ABI PRISM 3500xL Genetic Analyzer (Applied Biosystems, Waltham, MA, USA). Massive parallel sequencing was performed with Ampliseq technology on Ion S5 (Thermo Fisher Scientific, Waltham, MA, USA). We used the panel targeting coding exons of 52 genes that are associated with inherited diseases with cholestasis, including the *JAG1* and *NOTCH2* genes. Nucleotide variants were initially prioritized by minor allele frequency <2% in the general population and selected based on clinical diagnosis using human phenotype ontology (HPO) terms. Sanger sequencing was used to verify the mutation in the affected individuals and family members.

All founded genetic variants were interpreted using the ACMG/AMP Interpreting Sequence Variant Guidelines in accordance with the SVI Recommendations (https://www.clinicalgenome.org/working-groups/sequence-variant-interpretation/ (accessed on 10 July 2023).

### 4.6. Statistical Analysis

All results were expressed as *x ± s* in the figures as indicated. 

## 5. Conclusions

In our study, we analyzed the clinical and genetic characteristics of ALGS in pediatric patients with cholestatic liver disease. We have shown that the disease presentation is highly variable and that not all patients meet the classical diagnostic criteria. ALGS diagnosis may be especially difficult in infants with a few classical features or with a biopsy without bile duct paucity. Detection of pathogenic variants in the causative genes is particularly useful for patients who do not meet a sufficient number of clinical diagnostic criteria.

## Figures and Tables

**Figure 1 ijms-24-11758-f001:**
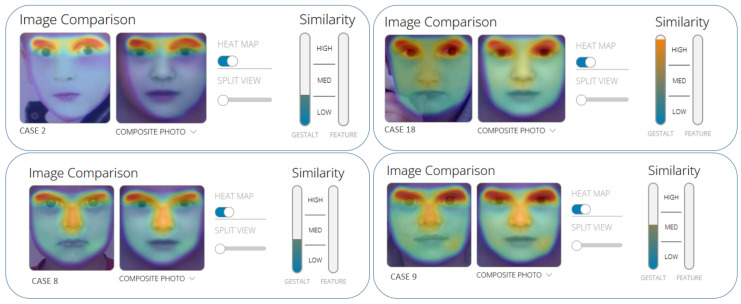
Face2Gene facial analysis in patients using (FDN Inc., Boston, MA, USA; https://www.face2gene.com, accessed on 10 July 2023).

**Figure 2 ijms-24-11758-f002:**
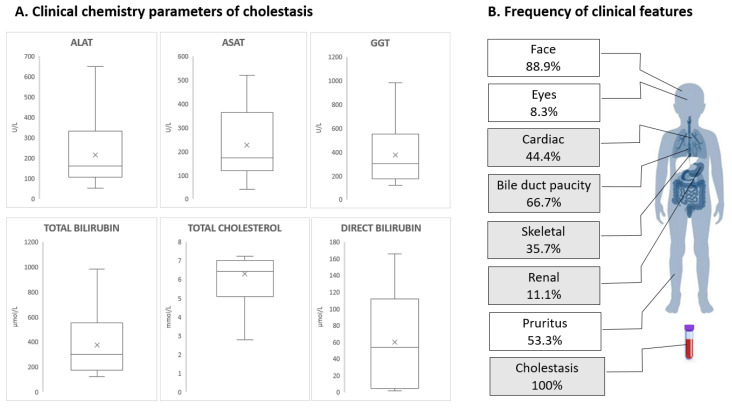
Biochemical and phenotypic characterization of the patients. (**A**) Laboratory characterization of cholestasis, including ASAT (normal range 0–40 U/L), ALAT (normal range 0–40 U/L), GGT (normal range < 34 U/L), total bilirubin (normal range 5–21 µmol/L), direct bilirubin (normal range < 3.4 µmol/L), and total cholesterol (normal range 3.2–5.2 mmol/L) using box plots. ASAT, aspartate transaminase; ALAT, alanine transaminase; GGT, gamma-glutamyl transferase. (**B**) Frequency of clinical features of ALGS patients.

**Figure 3 ijms-24-11758-f003:**
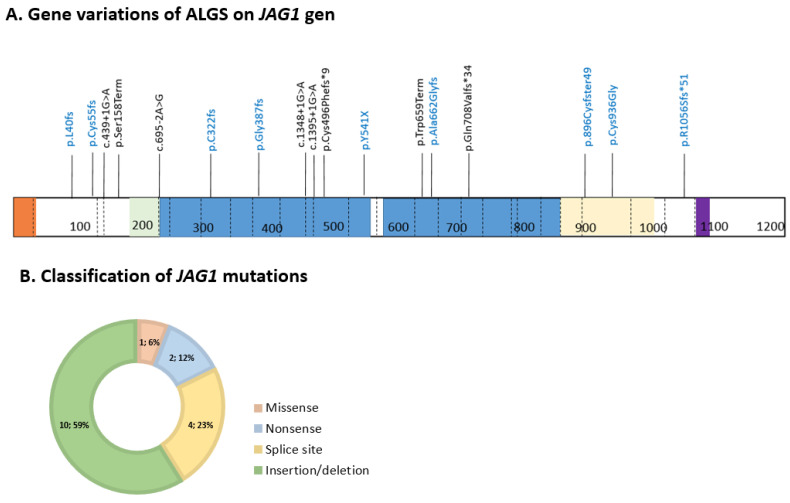
The range of identified variants in the *JAG1* gene in our cohort. (**A**) Gene variations of ALGS on the JAG1 protein: signal peptide (orange); DSL domain (green); EGF-like repeats (blue); cysteine-rich domain (yellow); and transmembrane domain (purple)—previously reported (black color) and novel (blue color) variants in *JAG1*. (**B**) Classification of *JAG1* pathogenic variants in the study cohort.

**Table 1 ijms-24-11758-t001:** Clinical characteristics and gene variations of ALGS on the *JAG1* gene.

N		Disease Features									Mutation Position (According to NM_000214.3) (Ref.)	Predicted Protein Consequence (According to NP_000205.1)	Segregation
		Cholestasis	Pruritus	Growth	Cardiovascular	Butterfly Vertebrae	Facial Features	Eye	Bile Duct Paucity /Age of Biopsy	Other			
1	f	+	-	−2.62	-	ND	+	ND	+/3 y	LT at 3 years of age	c.1619_1622dup	p.(Tyr541Ter)	de novo
2	M	+	-	−1.20	Atrial septal defect, PAS;	+	+	-	+/2 m	Necrotizing enterocolitis	c.1160del	p.(Gly387AlafsTer25)	inherited from unaffected mother
3	M	+	+	−1.74	Coarctation of aorta	+	+	ND	+/1 y	Renal cysts	c.2122_2125del [9]	p.(Gln708ValfsTer34)	de novo
4	M	+	+	−2.34	Ventricular septal defect, PAS	-	+	-	NA	Obstructive hydronephrosis	c.1976G > A [10,11]	p.(Trp659Ter)	de novo
5	M	+	-	−3.18	-	-	+	-	+/3 m	gallbladder hypoplasia	c.964dup	p.(Cys322LeufsTer5)	inherited from unaffected mother
6	f	+	+	−2.62	-	-	+	-	NA	Neonatal chylothorax	c.1395 + 1G > A [11,12]	p.?	de novo
7	f	+	-	−0.83	-	ND	+	ND	NA	Severe coagulopathy	c.120del	p.(Ser41ProfsTer5)	de novo
8	M	+	+	−0.60	-	-	+	-	-/2 m		c.1348 + 1G > A [13]	p.?	inherited from unaffected mother
9	f	+	+	−2.44	-	-	+	-	NA		c.1348 + 1G > A	p.?	inherited from unaffected mother
10	M	+	-	−3.01	-	-	-	-	-/2 m		c.2806T > G	p.(Cys936Gly)	de novo
11	M	+	-	−1.89	Bicuspid aortic valve; PVS	+	+	-	-/7 y	Development delay	c.1984dup	p.(Ala662GlyfsTer4)	adopted
12	f	+	NA	0.57	-	-	+	ND	+/3 m		c.1485_1486del [14,15]	p.(Cys496PhefsTer9)	de novo
13	M	+	+	−3.48	PVS	+	+	+	NA	LT at 1 year 3 months of age	c.3168_3169del	p.(Arg1056SerfsTer52)	de novo
14	M	+	NA	0.83	-	ND	-	ND	NA	gallbladder hypoplasia	c.2688del	p.(Trp896CysfsTer49)	de novo
15	M	+	-	−1.34	PAS	+	+	-	NA		c.695−2A > G [11,16]	p.?	de novo
16	M	+	NA	−1.62	PAS	-	+	ND	NA		c.473C > A [10,11]	p.(Ser158Ter)	de novo
17	f	+	+	−2.10	PAS	ND	+	-	+/5 y		c.164dup	p.(Cys55TrpfsTer18)	de novo
18	M	+	+	−1.59	-	-	+	-	NA		c.439 + 1G > A [11,17,18]	p.?	de novo

Note: “+”—positive; “-“—negative; M—male; f—female; ND—not determined; NA—not available; PAS—pulmonary artery stenosis; PVS—pulmonary valve stenosis; LT- liver; ?—This is HGVS-compliant variant descriptions on the protein level.

## Data Availability

The datasets used and/or analyzed during this study are available from the corresponding author upon reasonable request.

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
