# Peer review of "Clinical Characterization of Alagille Syndrome in Patients with Cholestatic Liver Disease"

_ijms, 2023, doi:10.3390/ijms241411758_

Round 1

Reviewer 1 Report

To the authors,

The article"Clinical characterization of Alagille syndrome in patients with cholestatic liver disease" is a clear manuscript. However, I have some questions and suggestions about it.

Question 1: Some of the reported patients inherit the mutation from their mothers. However, you do not provide any information about them. Are the mothers affected? Have they had any of the clinical characteristics of Alagille syndrome? Please clarify it.

Question 2: If the characteristic facial phenotype is the most common symptom of ALGS in your cohort, why do not you provide any picture of some of the patients? To present some of them would be very helpful, in my opinion.

Moreover, in last years several tools have been developed for the study of the facial characteristics of different genetic syndromes as FaceToGene, which are helpful to guide in the diagnosis of the genetic diseases. I strongly recommend using it with your cohort, specially in the patients that do not fit with the typical facial characteristics. The article Int J Mol Sci. 2020 Feb 4;21(3):1042. doi: 10.3390/ijms21031042 is an example of the use of these bioinformatic tools.

Question 3: Why do not you confirm pathogenicity of the missense variant c.2807T>G; p.Cys936Gly? You can do it in silico with several bioinformatic programs. I strongly recommend doing it. Moreover, you could provide as a supplementary information the chromatograms of the reported mutations you have found in your cohort.

Finally, there are some minor errors which should be corrected:

1. Define GGT in line 39.

2. You name first figure 1B in line 83 than figure 1A.

3. In table1 a novel column reporting if the patient fit the classical clinical criteria or not would be very informative.

4. You write fig 2B in line 140 but you refer to fig 1A

5. You write “Figure 1” in line 154 but you refer to Figure 2A

6. You do not name Figure 2B in the text, at least not correctly.

7. I find a bit confusing to put the Conclusion section after material and methods, I believe it would be more correct put it after the Discussion section.

Best regards

Author Response

First of all, we eager to thank you for detailed and thorough analysis of our manuscript as well as valuable comments and recommendations to improve it. Thereunder you could find responses to each comment.

Question 1: Some of the reported patients inherit the mutation from their mothers. However, you do not provide any information about them. Are the mothers affected? Have they had any of the clinical characteristics of Alagille syndrome? Please clarify it.

Answer 1: All mothers were unaffected (lines 164-171). Moreover, we added this information in to the table 1.

Question 2: If the characteristic facial phenotype is the most common symptom of ALGS in your cohort, why do not you provide any picture of some of the patients? To present some of them would be very helpful, in my opinion.

Moreover, in last years several tools have been developed for the study of the facial characteristics of different genetic syndromes as FaceToGene, which are helpful to guide in the diagnosis of the genetic diseases. I strongly recommend using it with your cohort, specially in the patients that do not fit with the typical facial characteristics. The article Int J Mol Sci. 2020 Feb 4;21(3):1042. doi: 10.3390/ijms21031042 is an example of the use of these bioinformatic tools.

Answer 2:

Thank you for your recommendations! We have added pictures, using Face2Gene application program. However, not all parents allowed us to use photos of their children.

Question 3: Why do not you confirm pathogenicity of the missense variant c.2807T>G; p.Cys936Gly? You can do it in silico with several bioinformatic programs. I strongly recommend doing it. Moreover, you could provide as a supplementary information the chromatograms of the reported mutations you have found in your cohort.

Answer 3:

We have added it in the table as a supplementary information.

We have provided as a supplementary information the chromatograms, but not for all patients. Some of them were lost.

Finally, there are some minor errors which should be corrected:

  1. Define GGT in line 39.

Answer:  We have changed the text accordingly.

  1. You name first figure 1B in line 83 than figure 1A.

Answer:  We have changed the text accordingly.

  1. In table1 a novel column reporting if the patient fit the classical clinical criteria or not would be very informative.

Answer:  We have added this information in text (line 194).

  1. You write fig 2B in line 140 but you refer to fig 1A

Answer:  We have changed the text accordingly.

  1. You write “Figure 1” in line 154 but you refer to Figure 2A

Answer:  We have changed the text accordingly.

  1. You do not name Figure 2B in the text, at least not correctly.

Answer:  We have changed the text accordingly.

  1. I find a bit confusing to put the Conclusion section after material and methods, I believe it would be more correct put it after the Discussion section.

Answer:  We agree with you. Initially, there was a different order of sections in the manuscript. After downloading the manuscript, the order of sections was changed. We not sure that we can change the order of sections.

Reviewer 2 Report

The presented paper is well written and of interest for the field. In the paper 18 cases of Alagille syndrome presented and analyzed regarding the clinical features and their genetics.

The paper as it stands is quite easy to understand there are a couple of clarifications left that would increase quality.

1.In the introduction it's written that 18 cases of Alagille syndrome were analyzed and they were diagnosed with Alagille syndrome. Further it's written that not all patients showed the classical features and met the diagnostic criteria. Therefore it would be of interest how the diagnosis was made in the first place.

2. Alagille syndrome is an autosomal dominant disease. Since some of the mutations are inherited from the mother of the children it would be of interest if the mothers were also affected.

3. Since only six of the 18 children displayed the classical features of the disease it would be of interest to know if some of the mutations that were analyzed are related to a less severe phenotype or especially in the inherited cases if there is a suspicion of interfering factors that might increase or decrease this is severity of the phenotype. A Comment on that would increase the value of the paper.

Author Response

First of all, we eager to thank you for detailed and thorough analysis of our manuscript as well as valuable comments and recommendations to improve it. Thereunder you could find responses to each comment.

  1. In the introduction it's written that 18 cases of Alagille syndrome were analyzed and they were diagnosed with Alagille syndrome. Further it's written that not all patients showed the classical features and met the diagnostic criteria. Therefore it would be of interest how the diagnosis was made in the first place.

Answer 1: All patient had cholestasis but not all had enough additional extrahepatic symptoms for making clinical diagnose Allagile syndrome. AS were just suspected and confirmed by DNA-test (we find pathogenic variants in the JAG1 gene). That why we concluded that DNA-test is very important in SA diagnostic.

  1. Alagille syndrome is an autosomal dominant disease. Since some of the mutations are inherited from the mother of the children it would be of interest if the mothers were also affected.

Answer 2: Yes, some of the mutations were inherited from mothers. All mothers were unaffected (we have added information in the text (line 167-174) and in the table). However, we don’t have information about childhood of this mothers. They could have transient neonatal cholestasis or something like this. Now, they have no symptoms of SA.

  1. Since only six of the 18 children displayed the classical features of the disease it would be of interest to know if some of the mutations that were analyzed are related to a less severe phenotype or especially in the inherited cases if there is a suspicion of interfering factors that might increase or decrease this is severity of the phenotype. A Comment on that would increase the value of the paper.

Answer 2: We have added information in the text (line 230-233).

Reviewer 3 Report

Dear Editor-In-Chief

Thank you for giving me the opportunity to revise the manuscript. 

Semenova et al., clinical and genetic characteristics of ALGS in 18 pediatric patients with cholestatic liver disease and introduced the genetic analysis importance in the disease identification.

The topic is interesting and the manuscript is well presented.

I recommend the authors to enrich the discussion part by incorporating other studies on the same topic like: https://doi.org/10.3390/livers2040021 and https://doi.org/10.1002/jgh3.12830 or other similar ones.

Moreover, in the introduction section, the priority of this kind of studies and also the strength of the present data must be included.

Author Response

First of all, we eager to thank you for detailed and thorough analysis of our manuscript as well as valuable comments and recommendations to improve it. Thereunder you could find responses to each comment.

I recommend the authors to enrich the discussion part by incorporating other studies on the same topic like: https://doi.org/10.3390/livers2040021 and https://doi.org/10.1002/jgh3.12830 or other similar ones.

Moreover, in the introduction section, the priority of this kind of studies and also the strength of the present data must be included.

Answer

We have enriched the discussion part accordingly recommendations (line234-243).
